

# Wormhole attack detection and mitigation model for Internet of Things and WSN using machine learning

Asma Hassan Alshehri

Department of Computer Science, College of Computer Engineering and Science, Prince Sattam bin Abdulaziz University, Alkharj, Saudi Arabia

## ABSTRACT

The Internet of Things (IoT) is revolutionizing diverse sectors like business, healthcare, and the military, but its widespread adoption has also led to significant security challenges. IoT networks, in particular, face increasing vulnerabilities due to the rapid proliferation of connected devices within smart infrastructures. Wireless sensor networks (WSNs) comprise software, gateways, and small sensors that wirelessly transmit and receive data. WSNs consist of two types of nodes: generic nodes with sensing capabilities and gateway nodes that manage data routing. These sensor nodes operate under constraints of limited battery power, storage capacity, and processing capabilities, exposing them to various threats, including wormhole attacks. This study focuses on detecting wormhole attacks by analyzing the connectivity details of network nodes. Machine learning (ML) techniques are proposed as effective solutions to address these modern challenges in wormhole attack detection within sensor networks. The base station employs two ML models, a support vector machine (SVM) and a deep neural network (DNN), to classify traffic data and identify malicious nodes in the network. The effectiveness of these algorithms is validated using traffic generated by the NS3.37 simulator and tested against real-world scenarios. Evaluation metrics such as average recall, false positive rates, latency, end-to-end delay, response time, throughput, energy consumption, and CPU utilization are used to assess the performance of the proposed models. Results indicate that the proposed model outperforms existing methods in terms of efficacy and efficiency.

## INTRODUCTION

Due to recent advancements in distributed computing and wireless transmission, *ad hoc* and wireless sensor networks are becoming more and more common. These types of networks are highly recommended in different applications like surveillance, security, and environmental monitoring, in the military, homes, and the healthcare industry (*Ali et al., 2018*; *Wang et al., 2022*; *Luo et al., 2022*; *Jiang et al., 2021*) technologies. Figure 1 shows a few applications of WSN networks, like WSN, a low-budget network, because it requires small sensing devices called sensors. These sensors can sense, process, and share the data among the other nodes in the network. These sensors also have a unique identity number and are

Corresponding author
Asma Hassan Alshehri,
asm.alshehri@psau.edu.sa

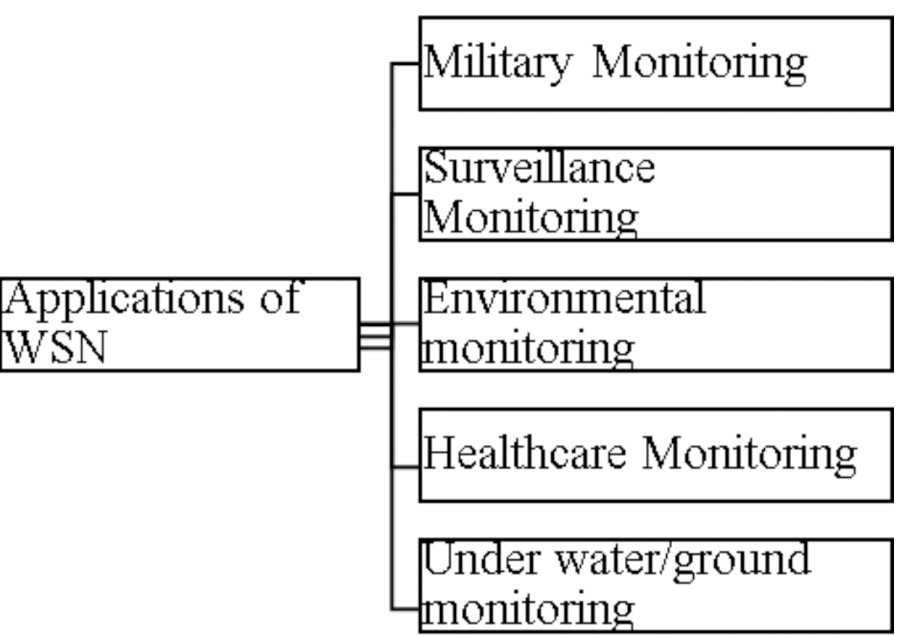

**Figure 1 Applications of WSN.**

easy to deploy. These sensors use wireless mediums for communication. The limitation of these sensors is that they cannot be protected using any conventional cryptographic algorithm (_Ramzan et al., 2022_; _Liu et al., 2022_; _Sun et al., 2018a_). These nodes use wireless mediums and are resource-constrained, so they must face different attacks. Wormhole attack is one of them. In a wormhole attack, a pair of malicious nodes (_Aqdus et al., 2023_; _Sun et al., 2019_; _Sun et al., 2018b_) become the authorized nodes of the network and then easily copy the important and secret data from that network. The wormhole attack was identified for the first time in _Zahra et al. (2022)_. To insert the wormhole in the network, a pair of malicious nodes becomes the authorized nodes of the network. As in Fig. 2, two different areas of the same network get these pairs of malicious nodes. After becoming the authorized nodes of the network, these nodes maintain a channel or link. The red lines in Fig. 2 represent these channels. These nodes also violate the currently running protocol of the network and start sending the data packets somewhere else through these channels. Other authorized network nodes consider these channels optimistic and send their confidential data packet through these corrupted links.

The WSN and IoT/cloud are now connecting for different benefits (_Hao et al., 2024_; _Xuemin et al., 2024_; _Bi et al., 2019_). IoT/cloud helps improve efficiency in everyday jobs. Daily, a huge amount of data is generated, and then the IoT or cloud provides a suitable path for data traveling. Using IoT, developer stores their valuable data and access it anytime. It also handles the problem without any delay. The transmission of packets from one place to another during a wormhole attack significantly disrupts the network (_Wang, Yang & Li, 2017_; _Zhang et al., 2023a_; _Zhang et al., 2023b_). Compared to standard routing, the tunnel has a greater capacity to transport packets. It has the potential to seriously harm the network

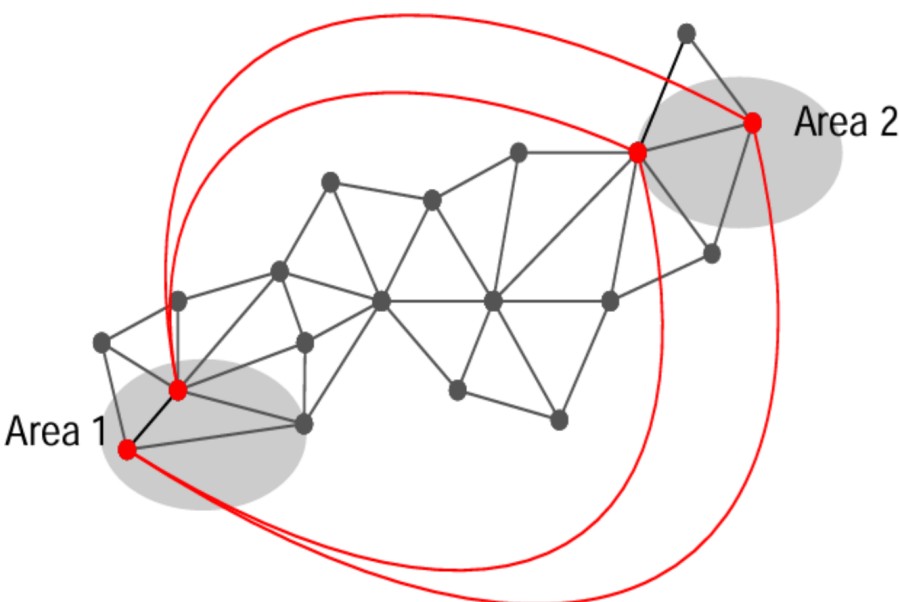

**Figure 2** **Wormhole attack in wireless sensor network.**

when combined with other assaults, such as a sinkhole attack. There are several ways that a wormhole attack can be carried out, including encapsulation, out-of-band/high-quality channel, and high-power transmission. Figure 3 demonstrates the integration of WSN in IoT. WSN is a technique that is used in every field nowadays, it is used for monitoring the domestic conditions of the home (*Almakdi et al., 2023*), to monitor and control the different parameters of the greenhouse (*Amin et al., 2021*) like temperature, humidity, water level, *etc*. Recent studies have shown that machine learning for IoT intrusion detection is expanding quickly. However, for threat detection on large-scale IoT networks, traditional machine-learning methods frequently exhibit low accuracy and/or reduced scalability. Even with current research efforts, anomaly detection by machine learning is still in its early phases. By focusing on deep-learning models for intrusion detection in an IoT setting, this article seeks to further this study. In this research, to detect the wormhole's malicious nodes, the connectivity information of the network will be used rather than any high computational algorithm (*Xie et al., 2024*; *Li et al., 2024*). This technique comes into existence after keenly examining the behavior of the wormhole links. A wormhole attack provides a new shortest path among the distant nodes of the network. This algorithm works by isolating the neighborhood of each node directly affected by the attack. The path size is different for the malicious node compared to the other nodes.

This study introduces a novel approach for detecting and mitigating wormhole attacks in IoT and WSNs by harnessing the inherent connectivity information within the network. Unlike conventional methods reliant on computationally intensive algorithms, our approach offers a streamlined solution that maximizes efficiency without compromising accuracy. We present a sophisticated hybrid detection model characterized by its scalability and efficiency, tailored to meet the demands of large-scale IoT and WSN infrastructures.

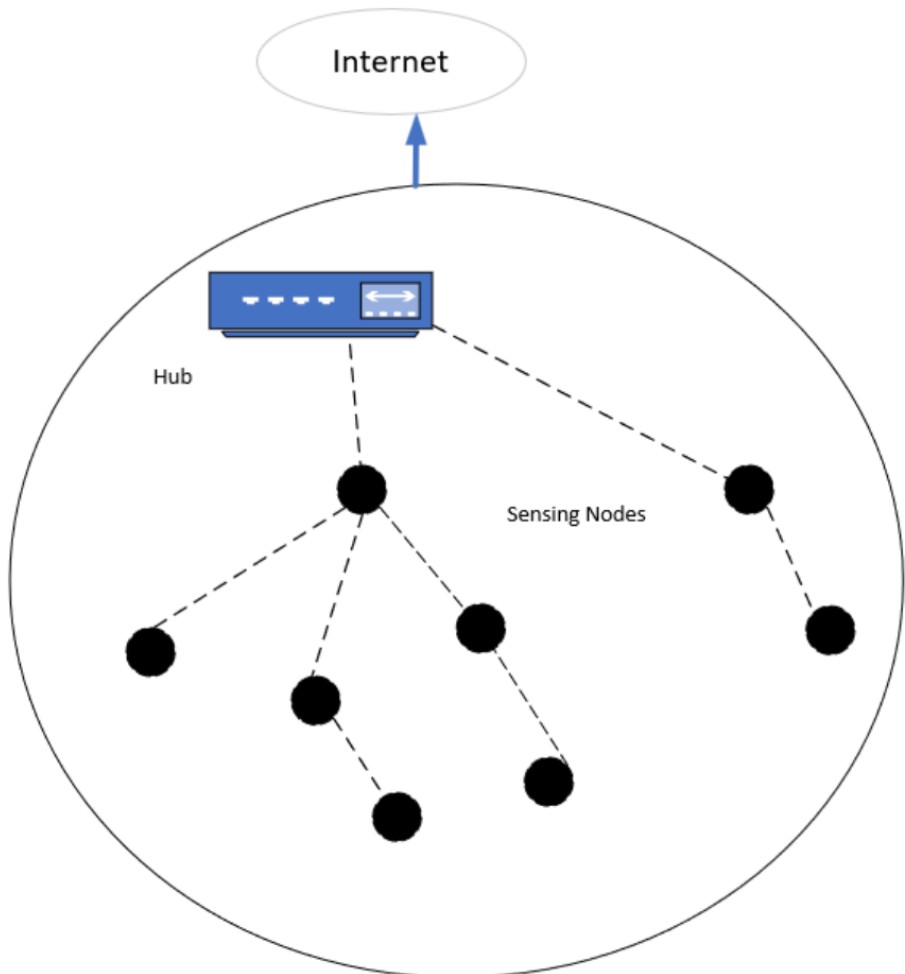

**Figure 3** **Integration of WSN in IOT.**

Through a judicious fusion of heuristic techniques and mathematical optimizations, our model achieves a delicate balance between computational complexity and detection accuracy, ensuring practical viability in real-world settings. Our contribution to this article is as follows:

⇒ To detect the malicious nodes in WSN, the connectivity information of the nodes in the network is utilized. We are identifying the useful dataset characteristics for the WSN and IOT framework for wormhole detection and isolation.

⇒ This research addresses the state-of-the-art issues with wormhole detection in wireless sensor networks and IoT, and AI and ML algorithms are recommended as the best approach.

⇒ A brand-new sophisticated hybrid method with polynomial complexity is created for wormhole isolation and detection. The algorithm is simple to use and does not require any special hardware.

⇒ The algorithm's theoretical detection probability is calculated. Simulation assesses performance regarding detection likelihood, network overhead, miss detection, and false node alarm rates. Results demonstrate the algorithm's effectiveness.

This article's further structure is as follows: the 'Related Work' enlists the related work. The 'Proposed Methodology' describes the proposed methodology. The 'Simulation Setup' illustrates the simulation setup. The 'Results and Discussions' elaborates on results and discussions. Conclusions have been drawn in the 'Conclusions'.

## RELATED WORK

In WSN, all the nodes have great security threats all the time. A wormhole attack is one of these threats. In the beginning, the problem of detecting wormhole attacks got the greatest attention of the researchers, which is why different protocols and countermeasures have been designed to detect wormholes in WSN. Different parameters like distance and time are used to solve the wormhole attack. In addition to addressing security issues with mobile *ad-hoc* networks (MANET), *Kouanou et al. (2024)* research suggests a novel machine learning-based approach for identifying and averting assaults. Using NetSim (Network Simulator) software, a 26-node MANET was created for the investigation. Wormhole and black hole assaults were then put into practice. Then, black hole and wormhole attacks were conducted. A machine-learning model was developed using a dataset generated from the network traffic gathered during the simulations to predict and recognize these attacks. *Ryu & Kim (2024)* Provide a brand-new multiple verification-based wormhole attack detection technique that uses these assaults' peculiarities. The suggested approach uses a trust mechanism to calculate each node's credit. Routing reduces the trustworthiness of suspicious nodes; those whose trustworthiness falls below a certain threshold are deemed evil. Based on federated deep learning and a dynamic trust factor (DTF), the suggested *Alghamdi & Bellaiche (2023)* effort proposes a cascaded wormhole detection approach for IoT networks. Although federated training guarantees data privacy and confidentiality at the node level for convolutional neural network (CNN) and long short term memory (LSTM) deep learning models, the DTF is predicated on two trust attributes.

*Patel & Patel (2016)* explained that two types of nodes exist in WSN: generic and gateway. For processing, sensing, and computing, multipurpose nodes are used and they are known as generic nodes, while for routing, gateway nodes are used. Every node in the WSN is assigned a random rank, and these nodes' rank is improved if they magnificently send the packet; otherwise, the rank is reduced. The adopted method (*Patel & Patel, 2018*) used round trip time (RTT) and hop count to detect the wormhole attack. Experimental results showed this technique has good accuracy in detecting the attack. The proposed method in *Harsányi, Kiss & Szirányi (2018)* detects and identifies the malicious nodes in the network. No special measurement was used here to detect and identify the affected nodes; only connectivity information was used. The adopted technique works in a distributed manner, and accuracy is not affected by the number of affected nodes in the network. In *Dwivedi, Sharma & Kumar (2018)*, the detection and prevention of wormhole attacks were revised by comparing different methods. Also, different models and modes of wormhole

attack were deliberated. In some techniques (*Khalil, Bagchi & Shroff, 2005*; *Khalil, Bagchi & Shroff, 2008*; *Poovendran & Lazos, 2007*) for malicious node detection, some particular nodes are also used. These nodes are known as guard nodes. These nodes know about their physical location and use their higher transmit power and different antenna appearances. The limitation of these techniques is that they are highly dependent on guard nodes. The proposed method in *Aliady & Al-Ahmadi (2019)*, the experiment was performed on network simulator 3 using ADOV routing protocol. Results came up with 100% accuracy in detecting the affected nodes in the network. No additional hardware was required to experiment, so the cost was also minimal. Wormhole attack, attack on the network layer and this attack had various modes however in (*Kumar Dwivedi, Sharma & Kumar, 2018*) high transmission power was considered and the received sign strength indicator (RSSI) was used to detect the affected nodes. In *As' Adi, Keshavarz-Haddad & Jamshidi (2018)*, a decentralized scheme is introduced in which the number of new neighbors and number of neighbors are considered as parameters. This scheme has low detection delay and traffic overhead. The experiment is performed on NS-3 and experimental results show high accuracy of detecting malicious nodes. However, all the authors try to detect the wormhole attack by using a complex algorithm. These algorithms required high computational time to solve the problem as the number of nodes increased. In our technique, only the information of the neighbor's nodes is enough to detect the malicious nodes in the networks, and it also requires less computation.

## PROPOSED METHODOLOGY

Our methodology has practical implications across various domains where IoT and WSNs are pivotal. In business, IoT devices manage inventory, logistics, and environmental monitoring, necessitating secure data transmission to protect operations and sensitive information. Healthcare relies on IoT for patient monitoring and telemedicine, demanding robust security against wormhole attacks to safeguard patient data and device reliability. In military applications, IoT enables battlefield surveillance and communication; our approach enhances network security to ensure secure communication channels in hostile environments. Smart cities utilize IoT for traffic management and public safety; our method strengthens network resilience against cyber threats, ensuring efficient city operations. By advancing ML-based detection techniques for wormhole attacks, our research secures IoT deployments across diverse applications, advancing wireless sensor network security. In our proposed solution, we leverage support vector machines (SVM) and deep neural networks (DNN) to address the challenges of detecting and mitigating wormhole attacks in wireless sensor networks (WSNs). SVM, a supervised learning model, is utilized for its effectiveness in binary classification tasks, enabling the identification of malicious nodes based on features extracted from network data. By training SVM on labeled datasets, it learns to distinguish between normal network behavior and anomalous patterns indicative of wormhole attacks. On the other hand, DNN, a powerful class of artificial neural networks, offers a more sophisticated approach to feature learning and abstraction, enabling the model to automatically extract hierarchical representations of network data. By leveraging

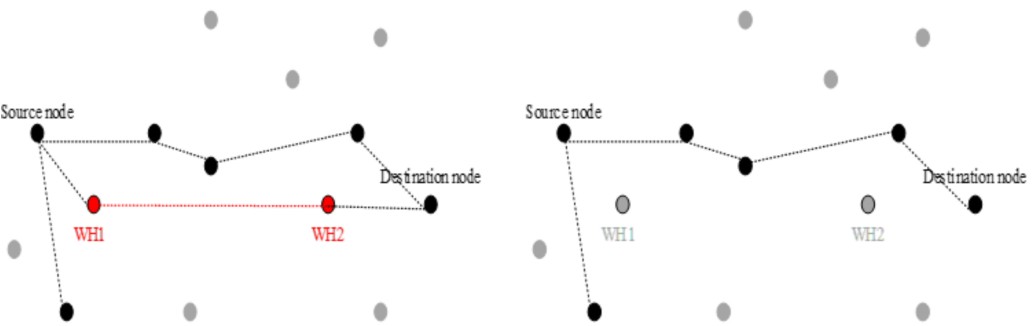

**Figure 4** **(Left) Network with wormhole attack.** The shortest path length between the source node and destination nodes is three hop size. (Right) Network without wormhole attack. The length of the shortest path between the source node and the destination node is four hop length.

the deep architecture of DNN, we aim to capture intricate patterns and relationships within the network data, enhancing the detection accuracy of wormhole attacks. Through a combination of SVM and DNN, our proposed solution seeks to provide a robust and comprehensive defense mechanism against evolving threats in WSNs, offering improved detection performance and resilience to sophisticated attack strategies. Many state-of-the-art methods have been proposed to deal with the problem of wormhole detection. However, all of these ended with some limitations. Some of these methods demanded special and expensive hardware or guard nodes. Our method identifies and isolates malicious nodes based merely on the connection information of the network nodes. The length of the shortest pathways connecting some of the sensing nodes was predicted to vary drastically with the removal of the wormhole channel. Some sensing nodes, however, will always have the same shortest route length. This assumption is shown in Fig. 4. As in Fig. 4A, the shortest path length from source to destination is three hop size. However, after removing the wormhole channel, the length of the new shortest path from source to destination becomes four hops, represented in Fig. 4B. In our article, we must examine the changes in the shortest path length, which is why we use a breadth-first search algorithm. To apply BFS in our graph, we randomly selected the source node. Breadth-first search sends the packet to its directly next hop. After receiving the packet, these nodes add one in the depth and forward it to its next hop directly. After receiving, they add one to its depth, and the same process is repeated until the destination is reached. Later when every node calculates its depth from source to destination, they send this information to its selected source node.

In our methodology, determining the optimal number of source nodes ('n') is crucial for the effectiveness of our approach in detecting wormhole attacks. We employ an iterative algorithm where initially, all network nodes are considered as potential source nodes. The algorithm prioritizes selecting the initial source node based on the smallest node identifier and then identifies and retains nodes within a certain hop distance ('n') from this source node. This iterative process refines the set of selected source nodes until no additional nodes can be added or until a predefined condition is met. The value of 'n' is strategically chosen to balance detection sensitivity with computational efficiency, considering factors such as

network size, density, and potential attack scenarios. In our proposed methodology, the determination of the smallest node ID, which serves as the initial source node, is crucial for the effectiveness of our wormhole attack detection strategy. We adopt an algorithm-based approach for assigning node IDs, ensuring a systematic and predictable distribution. Each node in the network is allocated an ID based on its geographical location and network topology. This location-based assignment allows for efficient and organized identification of nodes, facilitating seamless integration with our breadth-first search (BFS) algorithm for detecting wormhole attacks. By using a consistent and algorithmic ID assignment method, we can easily identify the smallest ID node, ensuring that our source node selection process is both logical and reproducible. This approach not only streamlines the initial selection but also enhances the overall accuracy and reliability of our wormhole detection mechanism, providing a robust defense against sophisticated attack strategies in wireless sensor networks. Implementation of this algorithm leverages a breadth-first search (BFS) strategy for efficient exploration of network connectivity, with parameters and heuristics fine-tuned through rigorous testing and validation to ensure robust performance across diverse network configurations.

## Selection of source nodes

The accuracy of our work is directly dependent on the number of source nodes. If we select a single node as a source node, the attack might be less affected by the picked node. So, that is why we try to select enough as source nodes. These nodes are also selected from different locations of the network because if the distance of the source node from both wormhole nodes is the same or too far from the source node, we cannot examine the changes caused by the removal of wormhole nodes. If we select many nodes as source nodes, it will increase the computational time. So, we use an algorithm to decide how many nodes are needed to be selected as source nodes. In that algorithm, all the nodes are initially considered to be the source nodes. Afterward, the nodes with the smallest node ID are selected from that set of expected source nodes as the first source node. For that time, this node behaves like the source node in the network and all the remaining nodes which are n-hop distance from the source are removed from the set of expected source nodes. This process iteratively continues until the set is not empty. According to this algorithm, the number of selected nodes directly depends on the value of n. If this value is small, too many nodes are selected as source nodes. However, if the value of n is large, the number of selected source nodes is small. Algorithm 1 outlines a method for selecting source nodes from a network based on a connectivity matrix and a specified value of 'n'. The algorithm initializes by setting the source nodes as all nodes in the network. Then, it iteratively selects the node with the smallest ID as the source node and identifies its 'n'-hop neighbors using breadth-first search (BFS). These neighbors are then removed from the set of source nodes. This process continues until all nodes have been considered as potential source nodes. When selecting the initial node, the decision is typically based on a predetermined criterion, such as the node with the smallest ID. Alternatively, other criteria such as node centrality measures or random selection may be employed depending on the specific requirements of the network and the application scenario.

---

**Algorithm 1:** Selection of source nodes from the network

Input: Network (in the form of connectivity matrix), value of n, set of all the nodes

Output: set of source nodes

*Step 1: initialization*

Source nodes set = set of all the nodes

*Step 2: selection*

Select the smallest ID node as the source node

Find the n-hop nodes for that source (using BFS.)

Remove them from the set of Source nodes set.

*Step 3: iteration*

Repeat the above step until the set of all the nodes is not empty

## Detecting the malicious nodes

We get the complete set of source nodes and search for the malicious nodes in the network. The first step is creating a matrix (size of matrix = no. of nodes in the network × no. of the source nodes). Then every node from the set of source nodes is picked and the following steps are performed on it. The distance of each source node from other nodes in the network is measured using a breadth-first search. These distances are stored in a vector. Now, we select a node from the set of all nodes, if that node is not in the set of source nodes as well as not even in the distance vector, we will perform the following steps to check the behavior of that node in a network as shown in Fig. 5.

- Analyze the data derived from network topology discovery statistically. If strange patterns appear, move on to step 2. Alternately, select a few routes for providing feedback to the source node.
- Send (test) data packets *via* the suspect pathways and watch for an ACK.
- If an attack is verified, alert the security legitimacy, the source, and/or the attackers' neighbours to isolate them from the network.
- The number of routes used in step 1 of routing protocols with numerous paths is a design parameter.

Although these paths depend on the multi-path data delivery strategy, maximum disjoint routes are preferred. Whether the hypothesized route is affected might be ascertained by the test in Step 2. When the attacker refuses to forward data packets but acts correctly during routing, it could be easy to identify a distinct kind of Denial-of-Service (DoS) attack. Step 3: To identify the malicious nodes, use the attack connection with the greatest relative frequency. The third step is very important and might be part of the signaling messages sent by an intrusion detection system (IDS) between local and global synchronized detection. These steps are performed for each network node, excluding the source nodes. After determining the distance for each node, the average variance is calculated individually. Then, the average of all these calculated variances is computed. Each node's variance is subsequently compared with this overall average. If a node's variance exceeds the computed average, it is identified as a candidate node. When we have a complete set of all the candidate

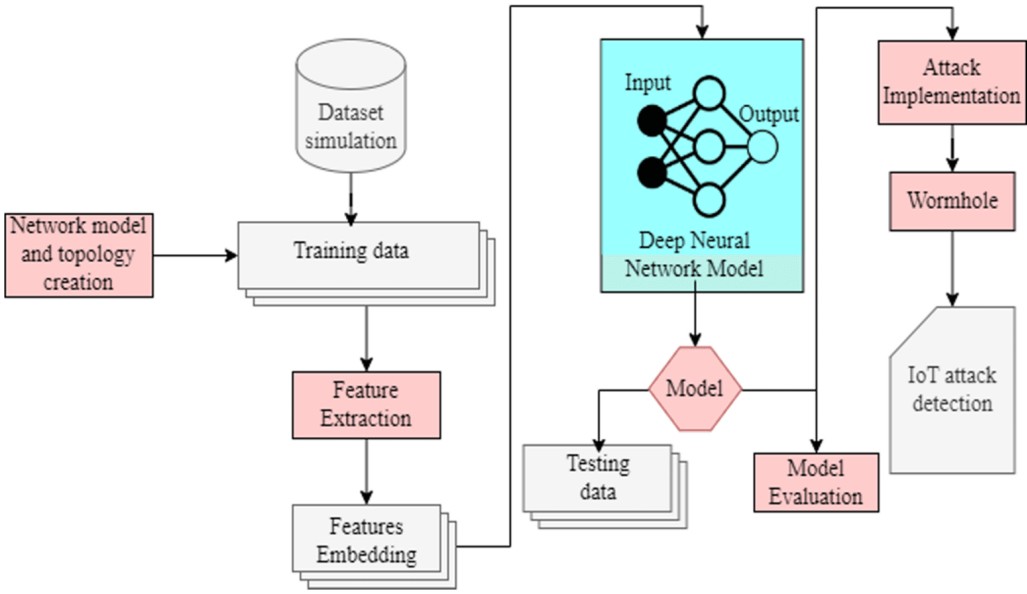

**Figure 5  Conceptual design and architecture of the proposed model.**

nodes, we will use a sub-graph inducted by these candidate nodes. If the inducted graph of any candidate node is of only one node, the node is not malicious. So, it must be removed from the candidate list. Afterward, we will close all the sensors other than candidate nodes. If they still communicate successfully its means they are malicious nodes. Extended malicious node detection and response in more detail are as follows:

### Fine-grained statistical analysis

Extend the statistical analysis in step 1 by incorporating fine-grained metrics, also as shown in Fig. 6. Explore features such as traffic patterns, node behavior, and communication anomalies.

### Traffic profiling for anomaly detection

Enhance the anomaly detection process in step 1 by implementing traffic profiling techniques. Create behavioral profiles for each node based on historical data and continuously update these profiles.

### Predictive analysis for path impact

Augment the impact analysis in step 2 by introducing predictive modeling. Utilize historical data and machine learning algorithms to predict the potential impact of a suspected path on network performance. This proactive approach allows preemptive measures to be taken before an attack occurs, minimizing potential disruptions.

### Adaptive route testing

Improve the route testing mechanism in step 2 by introducing adaptability. Dynamically adjust the frequency and intensity of route testing based on network conditions.

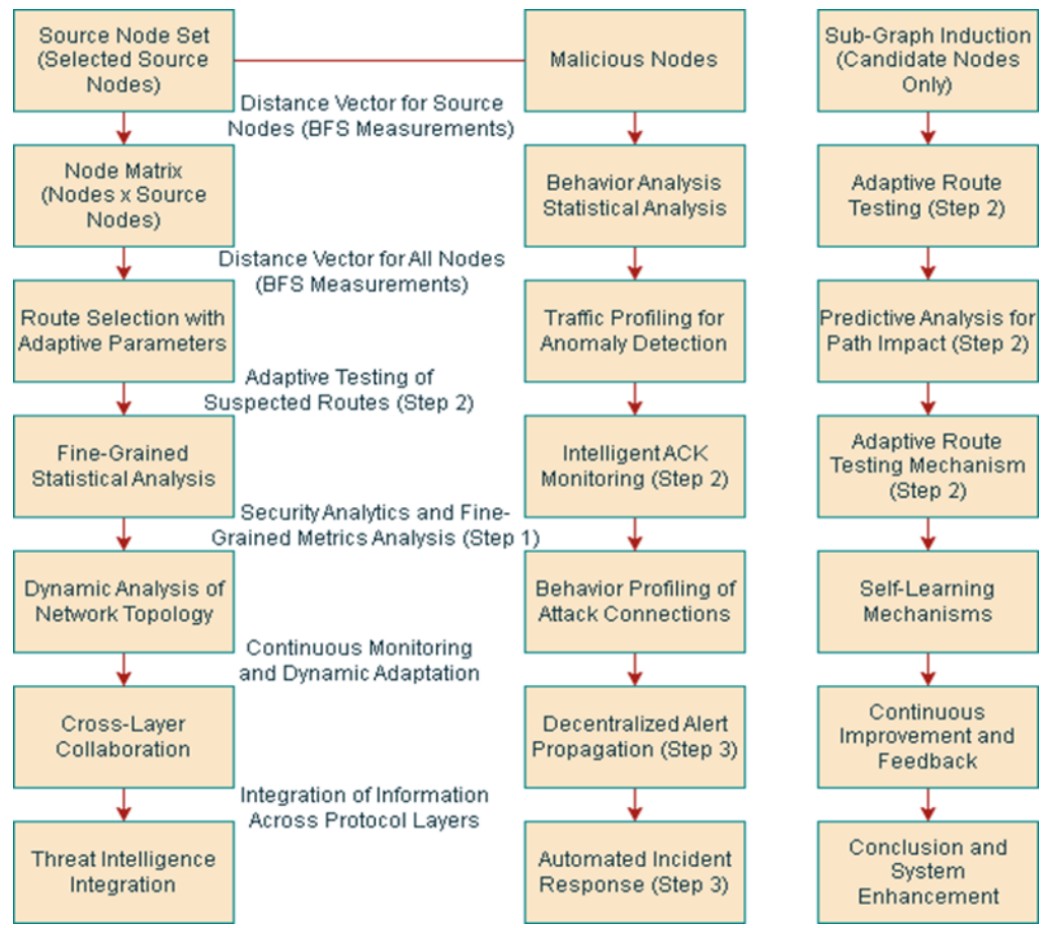

**Figure 6   A detailed diagram of proposed architecture through text description.**

### Behavior profiling of attack connections

Extend the analysis in step 3 by incorporating behavior profiling for identified attack connections. Create detailed profiles of attack connections, including traffic patterns, payload analysis, and temporal characteristics.

### Self-learning mechanisms

Implement self-learning mechanisms that allow the system to adapt to evolving threats continuously. The system can autonomously update its detection models based on real-time observations and feedback by leveraging machine learning algorithms.

## SIMULATION SETUP

In our experimentation, a laptop computer (Core i5, 8 GB RAM) and NS3.37 simulator were used to simulate the adopted technique to detect and isolate the wormhole attack in WSN. It is an effective tool for simulating mobile ad hoc networks and offers low-level analytical operations to analyze the network architecture, including sensor nodes, network links, application protocols, and queuing. We use different conditions to calculate the

performance of our result, like node development, communication models, and network density. The development models are random placement and a perturbed grid. Nodes in random placement are placed uniformly and independently within the given area. In the other model, the sensing nodes are perturbed from their initial locations (x, y) in a grid. This perturb is represented by the help of an equation, *i.e.,* [x-ab, x+ab] x [y-ab, y+ab]. Where 'a' is the displacement parameter and 'b' represents the square side length of the original grid. After deployment of these two models, the results show that random placement comes up with irregularities, while other models present good results. Two different types of graphs are used here to show the connections in the network. The first is a Unit Disk Graph (UDG) and the other is a quasi-UDG. In UDG the connection between the two nodes is established only when the distance between them is less than a communication radius. In the quasi-UDG model, the connection between two sensing nodes is established when the distance between them is less than some radius, also there exists another channel with some probability, *i.e.,* the distance between the radius and another radius that is equal to half of the first radius. The distance between the centers of the studied wormhole is also an important feature. If there are different end-nodes in the wormhole, that generates great distortion and increases the network's data traffic. Here, the wormholes are placed in such a way that in the original wireless sensor network, the set of wormholes are at a distance of at least eight hops. An average of false positives and recall are used to show the performance of this work.

In conducting our experiments, NS3.37 proved to be a valuable tool for modeling mobile *ad hoc* networks, offering detailed analytical capabilities to examine network components such as sensor nodes, links, and application protocols. We meticulously selected routing protocols to evaluate the network's efficacy and efficiency under various conditions, including detection and isolation of wormhole attacks. Our experimental setup encompassed diverse scenarios, including node deployment models (random placement and perturbed grid), communication models, and varying network densities. Providing details on data partitioning would involve explaining how the dataset was divided into training and testing sets, ensuring unbiased evaluation. Describing hyperparameter selection would entail clarifying the parameters chosen for algorithms or models and how they were optimized or tuned to achieve optimal performance.

## RESULTS AND DISCUSSIONS

To detect wormhole attacks in WSN and IoT only the connectivity information of the sensing node in the network is used. To implement this technique, no special and expensive hardware or guard nodes are required. The communication cost of each node in the network is also the same. Also, complex and expensive computations are essential for that detection. The accuracy of that technique is not affected by the number of nodes in the WSN. In Figs. 7 and 8 random placement UDG shows average numbers of false positives and average recall, along the *x*-axis average degree is represented and along the *y*-axis, the average number of false positives is represented. TF is a parameter that is the trade-off between the number of false positives and the detection rate. Here, three different values of TFs are used.

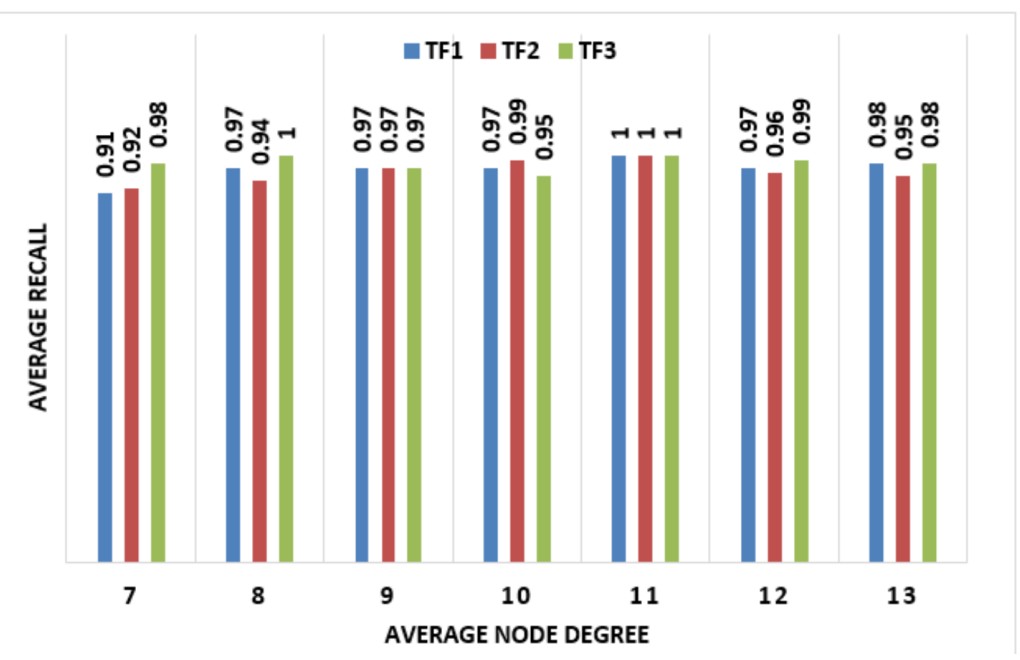

**Figure 7** Random placement UDG shows average numbers of false positives, along the *x*-axis average degree of nodes is represented and along the *y*-axis average number of false positives is represented. TF is a parameter that is the trade-off between the number of false positives and detection rate. Here three different values of TFs are used.

## Latency rate

The term "latency" refers to the amount of time required for an event to take place. The time it takes for data to go from location A to location B is sometimes expressed as a round-trip delay. The round-trip delay is crucial because a computer using a TCP/IP network only transmits a limited amount of data to its destination before waiting for an acknowledgment. The round-trip delay, therefore, has a big impact on network performance. Often, the delay is expressed in milliseconds (ms). The results show in Fig. 9 that even when a large volume of epoch's requests per second is met, there is only a tiny increase in overhead. The controller manages a large volume of requests per second, translating to a variety of service transmission rates. The system ought to be able to handle more requests if additional flows are used.

## End-to-end delay

IoT apps are used by systems that are constantly monitored thus it's crucial to do all activities as soon as feasible. That is why each cluster's head (CH) should be carefully selected. This issue is addressed by an algorithm that uses the Gdist distance measure to swiftly choose the CHs while considering the energy level of the sensors. A node is then marked when it is selected as the CH or when it is linked to another head. While choosing the cluster head, the CH must be chosen to reduce end-to-end latency. Figure 10 curves show the end-to-end latency as a function of simulation time for the proposed technique,

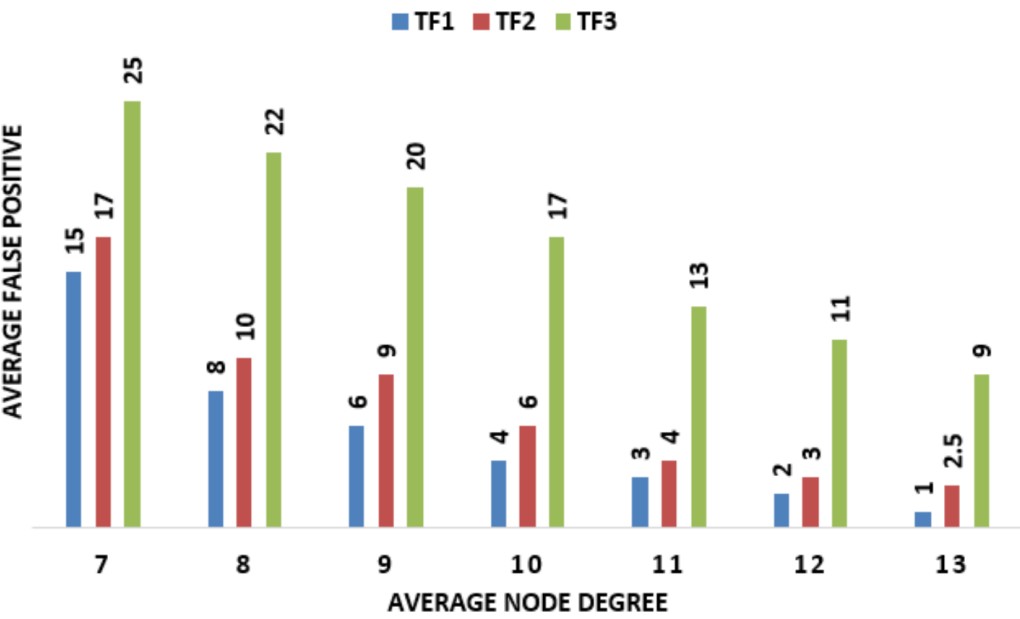

**Figure 8** **Random placements UDG through average recall are represented along the *x*-axis average degree of nodes and along the *y*-axis average recalls.** TF is a parameter that is the trade-off between the number of false positives and the detection rate.

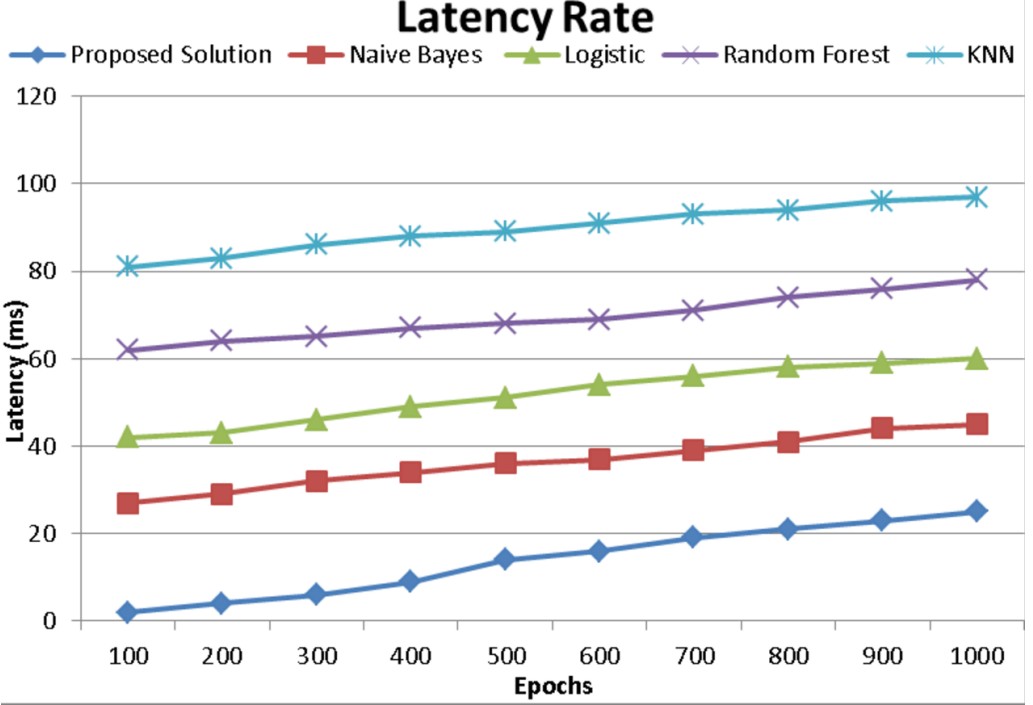

**Figure 9** **Latency rate comparison with existing techniques of proposed model.**

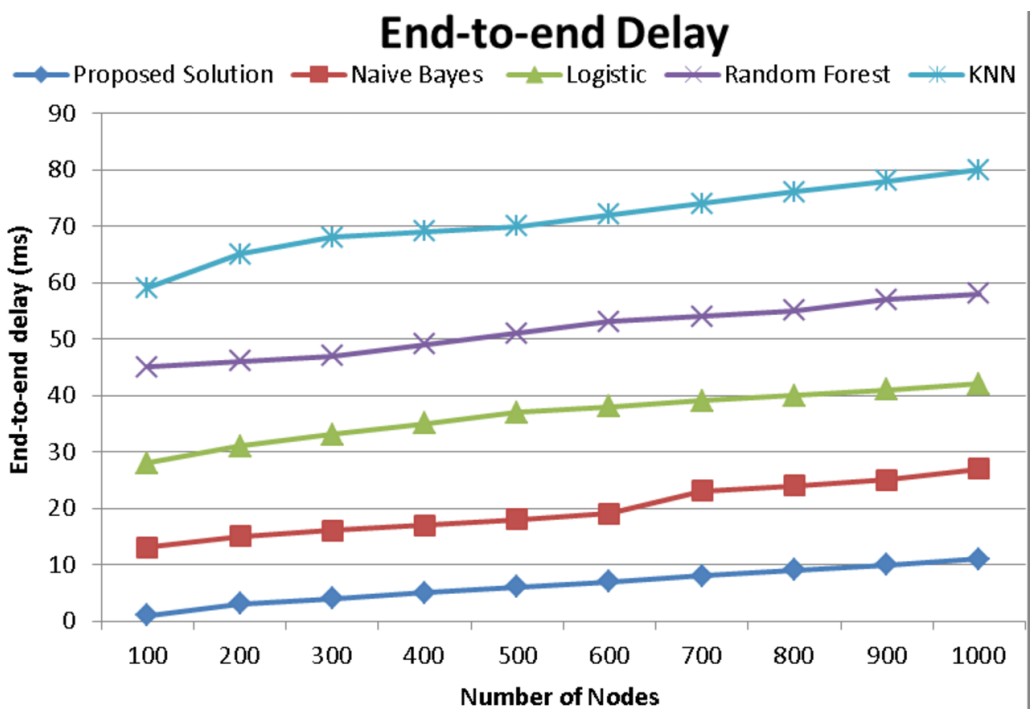

**Figure 10** End-to-end delay comparison with existing techniques of proposed model.

naïve Bayes, logistic, random forest, and KNN methods when they are run for seconds. Throughout the simulation process, the end-to-end delay of all techniques converges, however, the recommended way consistently has lower end-to-end latency than naïve Bayes, logistic, random forest, and KNN methods. As a consequence, our design provides sufficient performance to make the right CH selections and enable efficient communication among routing devices.

## Response time

It relates to the time it takes for data to be sent between two IoT devices. Our method is quicker than the traditional one given that the controller uses a proprietary routing protocol. Figure 11 displays the average response time for file transfers in different bulks across IOT nodes. Due to its reduced overhead, our solution outperforms the naïve Bayes, logistic, random forest, and KNN protocols. Figure 11 successfully depicts an efficiency analysis based on the number of nodes. Both response times increase as nodes increase in number. Additionally, when less frequent attacks are involved, it has been claimed that the suggested strategy performs better than the naïve Bayes, logistic, random forest, and KNN techniques. By leveraging the provided architecture of the cloud platform, all nodes get a quick response.

## Throughput

The suggested SVM with the DNN algorithm can recognize wormhole attacks and lessen their effects. Additionally, the throughput-based performance of the same network using

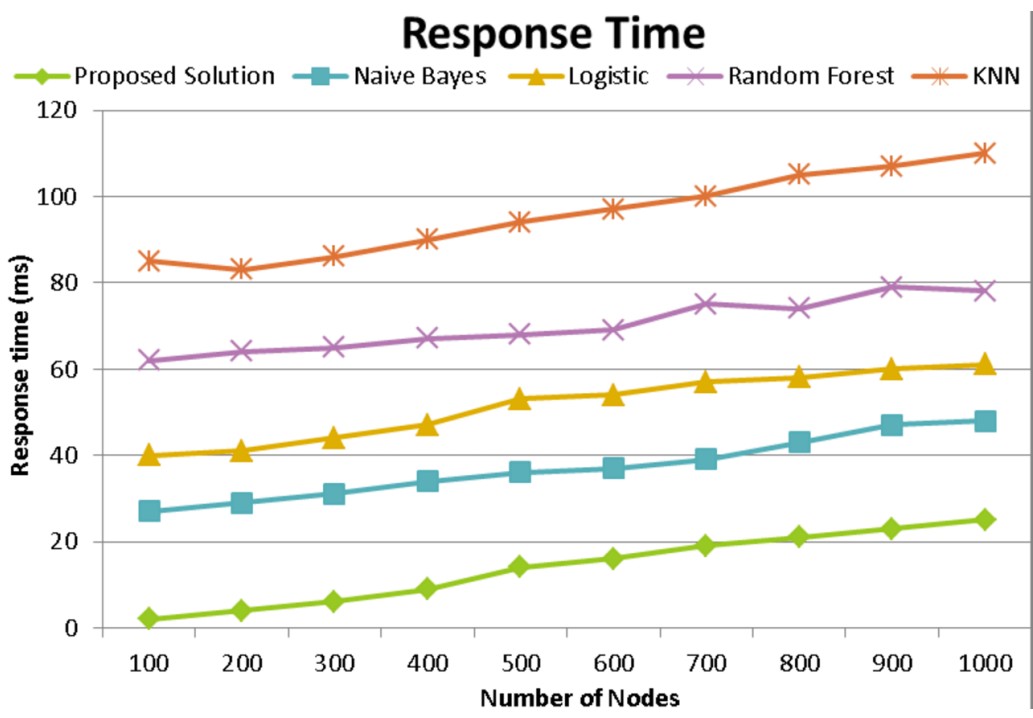

**Figure 11  Response time comparison with existing techniques of proposed model.**

both techniques is shown in Fig. 12. The network's characteristics and the number of nodes are the same as in the prior experiment. The capacity of the channel determines this. Throughput is often defined as the bandwidth used out of the allotted amount. The unit of measurement for network throughput is megabytes per second (MBPS). The performance of the suggested SVM with DNN is more acceptable based on experimental findings. The random forest-based strategy among them outperforms the two implemented methods.

## Energy consumption and CPU utilization

Energy consumption is the quantity of energy needed for various network functions. A joule measures how much energy is used by the proposed model, naïve Bayes, logistic, random forest, and KNN procedures. The proposed model is more secure and energy-efficient than the formerly supplied naïve Bayes, logistic, random forest, and KNN methods, as shown by the actual effects of nodes' energy usage.

It was also shown that the suggested approach effectively provides enough protection against these assaults by continuing the attack after a certain amount of time. The performance of the suggested SVM with DNN is shown to be more acceptable based on experimental findings than naïve Bayes, logistic, random forest, and KNN methods. While the proposed solution demonstrates higher CPU utilization, it concurrently exhibits lower energy consumption. This seeming inconsistency can be attributed to the intricate relationship between CPU utilization and energy consumption, influenced by factors such as algorithmic efficiency, computational complexity, and resource allocation strategies. The

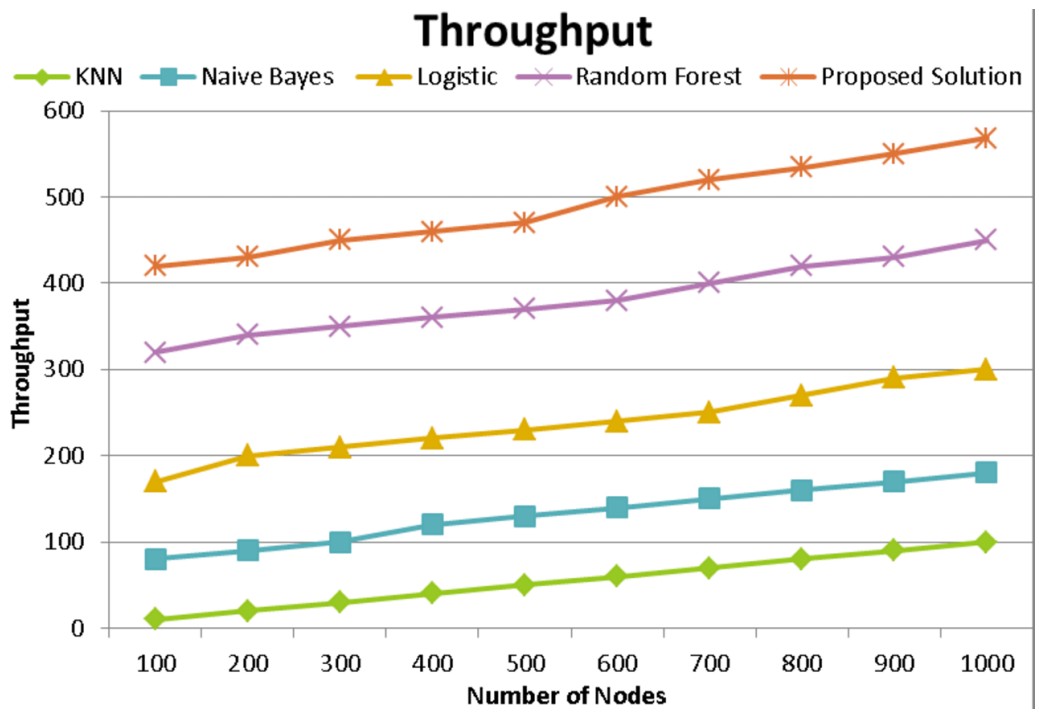

**Figure 12  Throughput comparison with existing techniques of proposed model.**

proposed solution may prioritize computational tasks that elevate CPU utilization while concurrently employing energy-efficient algorithms or resource management techniques to minimize overall energy consumption.

## Comparing the proposed solution with state-of-the-art methods

The proposed solution distinguishes itself from state-of-the-art methods by its reliance on existing network infrastructure, minimizing the need for specialized hardware or guard nodes. Unlike previous approaches that often make assumptions about network division or forged links, the proposed solution detects and isolates malicious nodes solely based on network node connectivity information. It employs a breadth-first search algorithm to analyze changes in shortest path lengths and dynamically adjusts the number of source nodes based on network topology and operational parameters. Moreover, the detection process integrates a comprehensive suite of advanced techniques including fine-grained statistical analysis, traffic profiling, predictive modeling, adaptive route testing, behavior profiling, self-learning mechanisms, decentralized alert propagation, cross-layer collaboration, and continuous improvement.

## LIMITATIONS AND FUTURE WORK

Despite the promising results achieved by our proposed methodology, several limitations warrant discussion. First, the detection accuracy of our approach heavily relies on the quality and representativeness of the training data used for the SVM and DNN. In real-world

deployments, acquiring extensive labeled datasets that capture all possible variations of normal and attack behaviors can be challenging. Moreover, the computational complexity associated with training and deploying DNN models could pose constraints on resource-limited WSNs, necessitating further optimization for efficiency. Another limitation lies in the assumption that node IDs are allocated based on geographical location and network topology. While this approach facilitates the identification of the smallest ID node, its practical implementation in dynamic and large-scale networks may face challenges such as frequent topology changes and node mobility. Furthermore, the BFS-based shortest path recalculations might become computationally expensive in very large networks, especially when real-time or near-real-time detection is required. Additionally, our methodology assumes a fixed communication radius for establishing connections, which may not account for variations in real-world environments where signal strength and connectivity can be influenced by factors such as interference and obstacles.

For future research, several directions can be explored to enhance the robustness and applicability of our proposed solution. One area of focus could be the integration of adaptive learning mechanisms that continuously update the detection models based on real-time network observations and feedback, thereby improving resilience to evolving attack patterns. Additionally, exploring the incorporation of cross-layer information could provide a more comprehensive view of network security, allowing for the detection of sophisticated attacks that span multiple protocol layers. Another promising direction is the investigation of decentralized and collaborative detection frameworks, where nodes share and verify alerts autonomously, reducing dependence on centralized entities and enhancing the robustness of the detection mechanism against network partitions and node compromises. Finally, extending the evaluation of our methodology to diverse and real-world network scenarios, including varying network sizes, densities, and mobility patterns, would provide valuable insights into its generalizability and practical effectiveness.

## CONCLUSION

Wormhole detection in WSN & IoT using machine learning was thoroughly studied and analyzed in this research conceptually and *via* simulation. This research suggested two strategies for identifying and avoiding wormhole attacks: hop-count analysis and specification-based intrusion detection. Wormhole attacks and suggested approaches were both simulated This study used the ML-based techniques as the best solutions for the highlighted state-of-the-art challenges in wormhole attack detection since they have a great deal of promise for controlling sensor networks efficiently. The base station employs machine learning models called SVM and DNN to categorize traffic data and identify malicious nodes in the network. This method is tested using traffic produced by the NS3.37 simulator in real-world environments. Evaluation metrics are used to assess the efficacy and efficiency of the suggested algorithms. These metrics include average recall and false positive rate, latency rate, end-to-end delay, response time, throughput, energy consumption, and CPU utilization. The results clearly show that the suggested model

performs better than the current approaches 93.12%, 89.34%, 87.34%, 79.67%, 81.94%, 72.90%, 75.76%, 84.56% and 90.32%, respectively.

### Funding
The authors received no funding for this work.

### Competing Interests
The authors declare there are no competing interests.

### Author Contributions
- Asma Hassan Alshehri conceived and designed the experiments, performed the experiments, analyzed the data, performed the computation work, prepared figures and/or tables, authored or reviewed drafts of the article, tools and methods evaluation, and approved the final draft.

### Data Availability
The Intrusion detection evaluation dataset is available at the University of New Brunswick: https://www.unb.ca/cic/datasets/ids-2017.html

### Supplemental Information
Supplemental information for this article can be found online at http://dx.doi.org/10.7717/peerj-cs.2257#supplemental-information.

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
