# Peer review of "Wormhole attack detection and mitigation model for Internet of Things and WSN using machine learning"

_PeerJ Computer Science, doi:10.7717/peerj-cs.2257_

## Round 0.1 · original submission · Major Revisions

The manuscript proposes a model for detecting and mitigating wormhole attacks in IoT and WSNs, but several areas require improvement. The introduction fails to sufficiently highlight the novelty and significance of the proposed model, lacking coherence in articulating its contributions. Additionally, the methodology for determining the number of nodes is unclear, hindering understanding. The writing quality needs enhancement for clarity and coherence. The experimental design lacks a comparison with state-of-the-art methods and detailed descriptions of the experimental setup, including data partitioning and evaluation metrics. Furthermore, the findings' validity is questioned due to unclear descriptions of methods and technical steps, as well as inconsistencies in figures and results. Specific improvements are needed, such as clarifying algorithm steps, revising confusing terminology, correcting references, and explaining conflicting results regarding CPU utilization and energy consumption. Addressing these issues is crucial for enhancing the manuscript's clarity, coherence, and validity of findings.

Reviewer 1 ·

Basic reporting

1. The contribution described in the Introduction lacks coherence and fails to sufficiently highlight innovation and key points. The introduction section should more coherently articulate the contributions of the paper, emphasizing the novelty and significance of the proposed model in detecting and mitigating wormhole attacks in IoT and WSN.
2. The methodology for determining the number of nodes, ‘n’, is not adequately explained. The technical description of the node selection process is fragmented, making it difficult to discern the primary focus of the method.
3. The writing quality needs improvement to enhance clarity and coherence.

Experimental design

1. A comparison with state-of-the-art methods is missing, which is essential to benchmark the performance of the proposed model.
2. The paper lacks a detailed description of the experimental setup, including the partitioning of data into training and testing sets, the selection of hyperparameters, and the definition of evaluation metrics.

Validity of the findings

1. The methods proposed are relatively conventional, and their descriptions are not sufficiently clear. For instance, the features fed into the SVM are not explicitly detailed.
2. The technical steps outlined from 3.2.1 to 3.2.9 lack clarity in terms of their objectives and contributions to the overall study.

Additional comments

1. The images, particularly Figures 2 through 5, are of poor quality and are blurry. Clearer figures are necessary for proper evaluation.
2. There are formatting issues throughout the paper, such as the incorrect formatting of section headings in Section 6 and the inconsistent use of italics for section 5.3. Additionally, there is a typographical error with the word “things” on line 17. A thorough proofreading is recommended.

Reviewer 2 ·

Basic reporting

This manuscript proposes to use merely distance information in network to detect wormhole attacks. It detects the attack by observing the number of hop differences, which is reasonable. However, some text needs improvement or fixes.

1. Line 52, fig.1 should be fig 2.
2. Some sentences are broken or hard to follow, requiring revision, e.g., lines 64, 76, 182, 193, 278.
3. I can not follow algorithm 1. When selecting the initial node in algorithm 1, how do you decide a node’s ID?
4. In section 3.2, some words are confusing. For example, “the average of all the averages“ and “average variance”. I would suggest revising the explanation of each step in this section.
5. Many references in the text do not match the figure, especially in the experiment part. Please correct them.

Experimental design

The experiment is well-designed and it performs the necessary measurement. There is something not clearly explained.

1. What does CH mean in 5.2?
2. In lines 298-299, the text refers to figures 7 and 8, which should be corrected with 8 and 9. What does the x-axis mean in Figures 8 and 9? I can not understand what a degree is.

Validity of the findings

1. In Figure 13, no proposed method is present.
2. In Figure 15, the proposed solution has the most CPU utilization but in Figure 14, it has the lowest energy consumption. Are the two results conflicted? More explanation should be given.

---

## Round 0.2 · Minor Revisions

Thanks for the revised manuscript. There are only some minor issues, and please fix them accordingly based on the reviews.

Reviewer 1 ·

Basic reporting

1. Highlight Practical Applications: The abstract mentions the application of IoT in various fields but does not illustrate how the proposed method applies to these areas or addresses specific issues. The authors should emphasize the potential impact of their method in real-world applications.
2. Optimize Sentence Structure: Some sentences in the abstract may be overly complex or verbose. The authors should simplify sentence structures to enhance clarity and conciseness.
3. Discuss Limitations and Future Work: If applicable, the authors should briefly mention any limitations of the study or directions for future research, which helps readers to understand the scope of the research comprehensively.

Experimental design

N/A

Validity of the findings

N/A

Additional comments

N/A

Reviewer 2 ·

Basic reporting

Thanks for the revised manuscript and it has been improved greatly. However, I still can not understand how to determine the smallest ID. Are the nodes pre-allocated with ID randomly or with an algorithm? It should be clarified.

Experimental design

The answer and revision addressed my concern.

Validity of the findings

The answer and revision addressed my concern.

---

## Round 0.3 · accepted · Accept

Thanks! The reviewer has no other comments.

Reviewer 1 ·

Basic reporting

I have no further commennts

Experimental design

N/A

Validity of the findings

N/A

Additional comments

N/A